# Promoting engagement with social fact-checks online: Investigating the roles of social connection and shared partisanship

Cameron Martel[1]*, Mohsen Mosleh[1,2], Dean Eckles[1,3], David G. Rand[1,3,4]

**1** Sloan School of Management, Massachusetts Institute of Technology, Cambridge, Massachusetts, United States of America, **2** Oxford Internet Institute, University of Oxford, Oxford, United Kingdom, **3** Institute for Data, Systems, and Society, Massachusetts Institute of Technology, Cambridge, Massachusetts, United States of America, **4** Department of Brain and Cognitive Sciences, Massachusetts Institute of Technology, Cambridge, Massachusetts, United States of America

☯ These authors contributed equally to this work.

* cmartel@mit.edu

**Data availability statement:** All analysis code and minimal anonymized data required for replicating our analyses are available on OSF at the following direct link: https://osf.io/nvk4u/.

## Abstract

Social corrections – where users correct each other – can help rectify inaccurate beliefs. However, social corrections are often ignored. Here we ask under what conditions social corrections promote engagement from corrected users, allowing for greater insight into how users respond to debunking messages (even if such responses are negative). Prior work suggests two key factors may help promote engagement with corrections – partisan alignment between users, and social connections between users. We investigate these factors here. First, we conducted a field experiment on Twitter (X) using human-looking bots to examine how shared partisanship and prior social connection affect correction engagement. We randomized whether our accounts identified as Democrat or Republican, and whether they followed Twitter users and liked three of their tweets before correcting them (creating a minimal social connection). We found that shared partisanship had no significant effect in the baseline (no social connection) condition. Interestingly, social connection increased engagement with corrections from co-partisans. Effects in the social counter-partisan condition were ambiguous. Follow-up survey experiments largely replicated these results and found evidence for a generalized norm of responding, wherein people feel more obligated to respond to people who follow them – even outside the context of misinformation correction. Our findings have important implications for increasing engagement with social corrections online.

## Introduction

There is widespread concern about the belief in and spread of false claims on social media [1–3]. One mechanism for attempting to reduce belief and sharing of false claims is social correction, wherein social media users correct one another. Numerous survey experiments

**Funding:** The authors gratefully acknowledge funding from Google through Google Research Scholar Award, gifts from Google to support other research, and funding via the National Science Foundation Graduate Research Fellowship under Grant No. 174530.

**Competing interests:** Other research by the authors is funded by Meta and Google. This does not alter our adherence to PLOS ONE policies on sharing data and materials.

indicate that corrections generally increase overall belief accuracy [4–6] while almost never backfiring by increasing belief in the debunked claim [7,8].

However, there is notably less work causally examining the effects of social corrections in the field. A key challenge for studying social corrections on actual social media platforms is that they are rarely engaged with. For instance, one observational study found that 73% of social corrections were ignored on Twitter (X) [9]. Given this constraint, researchers have largely focused on examining the downstream consequences of social corrections. For instance, observational analyses suggest that reshares of rumors on Facebook receiving a comment linking to a fact-checking article are more likely to be deleted - though large rumor cascades may continue to propagate despite the accumulation of such corrective comments [10]. And some experimental work suggests that corrections can negatively impact sharing quality, whereby users reshare more slanted and toxic content following receipt of a social correction [11]. However, given such low engagement rates, little work investigates how users directly respond to corrections they receive.

Promoting greater response rates to correction messages in the field is beneficial for several reasons. First, replies allow for examination of how the correction was received and provides insight into whether users are engaging in belief updating or resisting information after seeing a debunk message. Such direct feedback could then inform questions on correction efficacy and may elucidate whether specific aspects of correction messages are particularly effective or ineffective [12–14]. Second, while direct responses to corrections may be positive or negative, engagement itself may be a beneficial and constructive outcome of corrections. Engagement and reciprocal communication have been posited as necessary aspects of deliberative discourse [15,16] – and even when engagement is initially negative or challenges the correction, such engagement initiates a dialogue that may ultimately open the door to belief updating. Typical one-off approaches for correcting misinformation have reliable but ultimately small and limited effects in magnitude and scope [17,18], and so rebuttal mechanisms that elicit engagement over non-responses may be more effective at promoting sustained learning and belief change [19,20].

Social corrections that promote engagement help us learn more about correction efficacy and may initiate beneficial dialogues conducive to downstream belief change. Here, we use a combination of field and survey experiments to shed new light on social correction by causally investigating what factors promote this crucial first step of initiating preliminary engagement with correction messages.

Our experiments focus on two factors which prior work suggest may be important for eliciting engagement with corrections: partisan alignment between the corrector and the corrected user, and social connection between the corrector and the corrected user. Evidence suggests that partisanship may impact how individuals respond to corrections. For example, one survey experiment found that corrections from Republicans were more effective at correcting climate misinformation believed by fellow Republicans [21]. Other research, however, suggests that debunking messages from unlikely partisan sources may be more impactful upon both co-partisans and counter-partisans [22] or that partisan source may have little effect on belief updating [23]. It is important to experimentally investigate whether shared partisanship is a pre-requisite for engagement with social corrections online.

Separately, observational research suggests that Twitter users are more likely to respond to corrections [9], and to have responses which are positive and indicative of correction acceptance [24], when the correction comes from a user with whom they have social connections (e.g., both corrector and corrected user follow each other). However, since this prior work is observational, it remains unclear whether this effect is causal. Furthermore, even if it is

causal, it is unclear whether it is necessary for there to be an actual pre-existing offline relationship, or if the kind of minimal virtual social connection that social media can facilitate (e.g., 'following' someone, engaging with their posts) is sufficient to increase engagement with corrections.

What are the effects of partisan alignment and virtual social connection on engagement with corrections? Here, we experimentally test this question. First, we conducted a randomized field experiment on Twitter (now X; N = 1,586 corrections sent to unique users; 1,456 corrections successfully delivered) in which we used human-looking bot accounts to deliver corrections to users who shared articles fact-checked as false by a professional third-party fact-checking organization. We varied whether our corrector bots identified as Democrats or Republicans, and whether they followed the user and liked three of their tweets the previous day (thus creating a minimal social relationship). We then assessed whether users engaged with the public correction they received. We also offer preliminary evidence as to whether such initial engagement was positive or negative.

Next, we report the results of a follow-up survey experiment conducted on Amazon Mechanical Turk (N = 808) in which we aimed to replicate our field experiment findings. We also conducted another survey experiment (N = 1,606) examining the role of communicative reciprocity norms on the relationship between minimal social connections and engagement with both corrective and non-corrective messages in order to gain mechanistic insight into why individuals respond to corrections.

Across both our field and survey experiments, we also examine partisan extremity as a potential moderator of our social connection and shared partisanship treatment conditions on correction engagement.

## Field experiment

### Methods

To examine what corrector account features promote engagement with social corrections, we conducted a field experiment on Twitter (X) in which we identified users who shared links to fact-checked false political articles and replied to their tweets with public corrective messages from human-appearing bot accounts. Our corrective messages contained short messages questioning the veracity of the false story claim and provided a link to a fact-checking article from Snopes.com.

**Participants.** We identified Twitter (X) users who had written posts containing links to one of 11 political articles, varying in their appeal to liberals and conservatives, rated false by the fact-checking website Snopes.com in the two months prior to launching our Twitter field experiment (11-November-2019 to 3-January-2020; see S62 Table in S1 Text for list of stories). For each user who shared one of these links, we collected general account information (e.g., number of followers, followed accounts, tweets) and estimated political ideology using the accounts they followed [25]. As pre-registered we removed 'high status' users (users with over 10,000 followers) and users for whom we could not estimate ideology because they did not follow any political accounts. We then selected 2,000 users for inclusion in our study to maximize article political balance – this led to 24.6% of sampled users estimated as liberal, and 75.4% estimated as conservative.

**Procedure.** We created six identical Democrat bots and six identical Republican bots, only varying by the American male account name. Bots indicated partisanship via their profile names and bios. To assess whether users attended to our bot account partisanship, we evaluated the rate at which users followed-back our accounts in the conditions where users were followed before receiving a correction. Consistent with prior research [26–28], we found

that users were significantly more likely to follow-back co-partisan versus counter-partisan accounts – providing evidence that users in our study were indeed attending to the portrayed partisanship of our bot accounts (see Sect 4h in S1 Text). To appear authentic, we created the accounts roughly 3 months before interacting with users. As per [29], we also purchased each account 1,000 followers, to further increase authenticity. Before interacting with users, each account followed a set of major political figures and mainstream news outlets aligned with their ideology [25] and randomly retweeted about 50 recent tweets from those accounts. These retweets continued during the course of the experiment.

We also varied whether each account followed and liked the three most recent tweets of the to-be-corrected users one day prior to sending correction replies – thus establishing a minimal social connection. The primary component of this social connection is the unilateral tie created by our following the targeted account. Liking three recent tweets was included as part of this treatment in order to increase the perceived authenticity of our accounts, as well as increase the likelihood that users saw and remembered notifications from our account.

We performed randomized assignment by blocking [30] by creating relatively homogenous blocks of users based on (i) number of followers, (ii) number of tweets in the past two weeks, (iii) number of mentions in the past two weeks, (iv) number of replies in the past two weeks, and (v) political ideology. We then randomly assigned each set of blocks to a treatment day to have a balanced number of users across experimental conditions receive corrections each day.

For the correction reply itself, although research has found no significant effect of the tone of corrective messages on intentions to reply versus ignore [12–14] nor substantial differences in the sophistication or formatting of corrections [31,32], out of caution we used non-confrontational language in our corrections (see S63 Table in S1 Text for all 12 variations of the corrective messages we used) in a standardized format. To reduce the chance of corrected users noticing several similar corrective messages in the "tweets and replies" section of the bot profile, each bot would randomly retweet from the elite accounts it followed after sending each corrective reply.

**Open science.**   We pre-registered our field experiment here. We note two important deviations: (i) we were unable to send 2,000 corrections because of interference from Twitter security, and (ii) we observed unexpected differential treatment delivery failure across conditions, resulting in our pre-registration being under-specified as to how to address participants who did not receive full treatment (therefore, while we attempt to adhere to our pre-registration as closely as possible given these constraints, our field experiment analyses should not be considered pre-registered). We report additional robustness analyses using varying inclusion criteria in our SI. All analysis code and minimal anonymized data required for replicating our primary field analyses are available here (https://osf.io/nvk4u/).

**Ethics.**   Our experimental setup was approved by the MIT Committee on the Use of Humans as Experimental Subjects (COUHES) protocol #1907910465. We received a waiver of informed consent as this was essential for the experiment to have ecological validity (i.e., for participants to not know they were part of an experiment), and the experiment posed no more than minimal risk to all persons. Minimal risk is typically evaluated by comparison with the risk people would be exposed to otherwise; being misinformed about a Twitter (X) account's operator frequently occurs in the absence of our intervention. Furthermore, we did not debrief users. This is because we were unable to send private messages to the users, because they did not follow our accounts. We believed that publicly informing users (and their followers) that they had been part of an experiment on false news sharing would be a violation of their privacy, and that this would be a greater harm than not debriefing them given the minimal deception - the corrective tweet content was entirely accurate, the only

deception was not identifying the corrector as a bot (which was needed in order to study social corrections).

## Results

As outcome variables, we assess (i) whether the user engaged with our corrective message (i.e., whether the user replied to, retweeted, or favorited our corrective message, or deleted the tweet containing false news), (ii) whether the user engaged positively with our corrective message (belief updating, liking/retweeting the correction, deleting the original false news tweet), and (iii) whether the user engaged negatively with our corrective message (resisting corrective information). All Twitter (X) field data were collected using the (then existing) Academic Twitter API.

Positive and negative engagement were then determined via ratings from a separate Amazon Mechanical Turk (N = 66) survey in which workers were asked to rate correction replies as either 'belief updating' or 'resisting information' (see Sect 4f in S1 Text). We also counted instances where the user favorited or retweeted the correction or deleted the original tweet containing false news as positive engagement.

Unexpectedly, the fraction of users to whom corrections were not successfully sent varied across experimental conditions (e.g., error indicating that tweet to reply to was "deleted or not visible to you"; baseline co-partisan: 3.79%, baseline counter-partisan: 4.50%, social co-partisan: 6.43%, social counter-partisan: 17.96%; Fisher's exact test, p < 0.001). The message failure rate was specifically substantially higher in the social counter-partisan condition, likely driven in large part by users blocking the counter-partisan bot after it followed them (subsequent Twitter (X) field research has indeed found that users are 12 times more likely to block counter-partisan accounts compared to co-partisan accounts [33]). The message failure rates across the three other conditions were not significantly different (Fisher's exact test, p = 0.224). This differential message failure rate creates an unforeseen challenge for making causal inferences about the differences in engagement involving the social counter-partisan condition. No such challenge exists, however, when making comparisons between the other three conditions (as message delivery failure rates did not differ). Therefore, we first consider models only including the three conditions for which the error rate was not significantly unbalanced (i.e., omitting N = 329 in the social counter-partisan condition).

Among users in the co-partisan condition (N = 745), we find via a linear probability model (as pre-registered; linear models used as primary analyses in the current work unless stated otherwise) a significant positive effect of social connection on engagement (b = 0.066, SE = 0.032, t(743) = 2.03, p = 0.042; See S69 and S76 Tables in S1 Text), such that being followed by a co-partisan and having them like previous tweets increased the user's likelihood of engaging with the correction. Minimal social connection increases correction engagement amongst co-partisans.

Interestingly, we found that in the co-partisan condition, social connection significantly increased negative engagement (b = 0.052, SE = 0.026, t(743) = 2.02, p = 0.043; See S70 and S77 Tables in S1 Text) and had a directionally positive but non-significant effect on positive engagement (b = 0.014, SE = 0.024, t(743) = 0.57, p = 0.571, 95% credible interval = [-0.03, 0.06] using Bayesian regression analysis [throughout the current work, we perform exploratory Bayesian regression analyses using weakly-informative normal priors over intercept and effect of predictors and a weakly informative prior over residual standard deviation to provide additional information around our estimation of non-statistically significant results]; See S71 and S78 Tables in S1 Text); however, these two effects were not significantly different from one another (p = 0.273 via z-test). This means that we do not find

evidence that the social condition increased negative engagement more than it increased positive engagement (conditional on co-partisanship) – although due to limited power we do not precisely estimate whether there is evidence for no difference between effects on negative versus positive engagement. Altogether, the results remain opaque as to whether social connection increases negative engagement in particular, or increases both negative and positive engagement.

Among users in the baseline (non-social) condition (N = 763), we do not find a significant effect of shared partisanship on engagement (b = 0.003, SE = 0.031, t(761) = 0.11, p = 0.916; 95% credible interval = [-0.06, 0.06] using Bayesian regression analysis; See S72 and S79 Tables in S1 Text), such that users were not significantly more likely to engage with corrections from co-partisans compared to counter-partisans. We also did not find a significant effect of shared partisanship in the baseline condition on either positive engagement (b = 0.000, SE = 0.023, t(761) = 0.01, p = 0.990; 95% credible interval = [-0.05, 0.05] using Bayesian regression analysis; See S74 and S80 Tables in S1 Text) or negative engagement (b = 0.003, SE = 0.023, t(761) = 0.901, p = 0.901; 95% credible interval [-0.04, 0.04] using Bayesian regression analysis; See S73 and S81 Tables in S1 Text).

Importantly, the social connection effect in the co-partisan condition was significantly larger than the shared partisanship effect in the baseline condition (Wald test, $\chi2(1) = 4.70$, p = 0.030), providing some evidence that prior social connection has a larger effect on promoting engagement with corrections than shared partisanship.

We next examine what happens when including data from the social counter-partisan condition. To do so, we address differential message delivery failure rates across conditions via a principal stratification approach [34,35]. This allows us to estimate the effects of our experimental conditions among users for whom the correction was successfully delivered by leveraging available information about users for whom the correction was not successfully delivered. This principal stratification approach necessitates additional assumptions, including: (i) complete randomization of treatment assignment (with an intermediate variable – i.e., successful treatment delivery), (ii) monotonicity (i.e., given that people are less likely to receive treatment in the social counter-partisan condition, any user receiving treatment in that condition would also have received treatment if assigned to one of the three other conditions), (iii) an exclusion restriction (i.e., complete mediation, such that predictors of correction delivery only affect engagement through treatment delivery, not directly), and (iv) assuming principal ignorability (i.e., potential outcomes are independent of successful treatment delivery given observed predictor covariates; see [34]).

We first predicted the probability of successful correction delivery for all users in our three unaffected conditions using the following predictors: We trained a random forest classifier predicting whether the intervention was successfully delivered to the user using user partisanship, partisanship extremity, total number of replies, number of replies in the past two weeks, total number of tweets, number of lists, number of followings, number of followers, and number of likes. We chose to use a random forest classifier for this prediction task given research demonstrating random forest classification can be a highly accurate and effective classification procedure [36] – and the better we can predict successful intervention delivery for users, the better our principal stratification approach can help address differential message delivery failure rates across conditions. We then re-ran our analysis excluding users for whom the corrections failed (including only N = 1,456) while additionally weighting observations by each user's predicted probability of successful correction delivery had they been assigned to the social counter-partisan condition. This helps us to address the causal inference challenge arising from our inability to know which users in the other three conditions would have blocked us had they been assigned to the social-discordant condition.

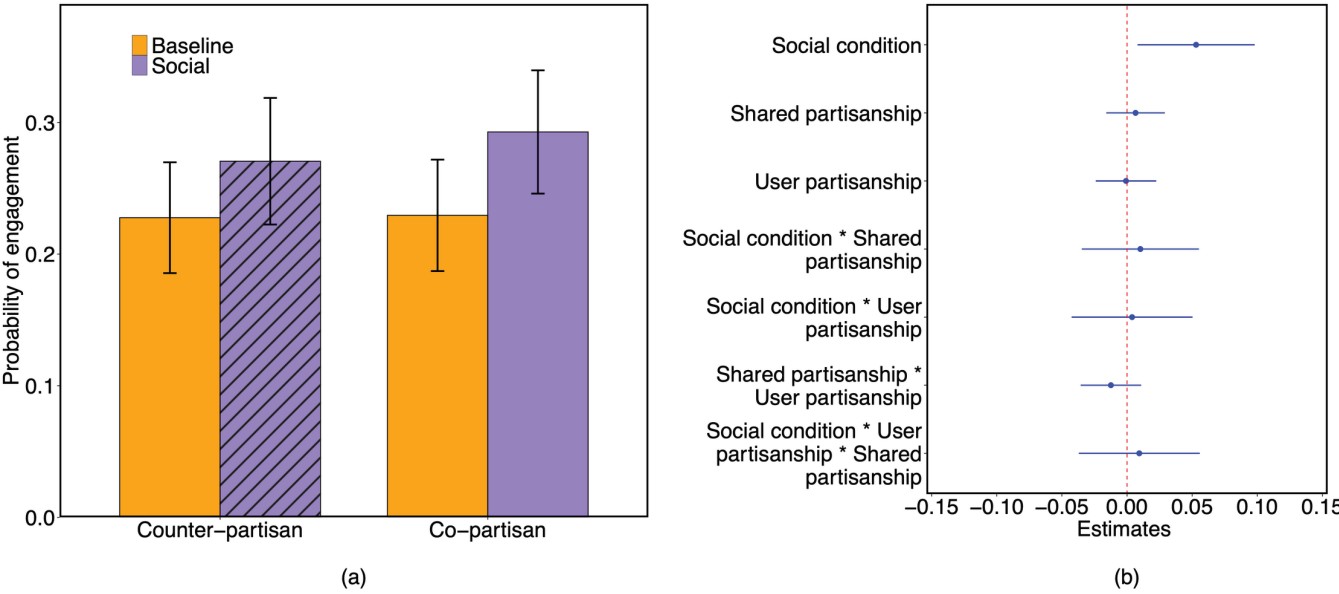

**Fig 1. Prior social connection increases engagement with corrections in our Twitter field experiment, particularly given shared partisanship.** Probability of engagement with the corrective message in the field experiment is shown across experimental conditions. Panels (a) and (b) show, respectively, the probability of engagement and effect sizes from the linear model when weighting observations in conditions other than the social counter-partisan condition (indicated by diagonal lines) based on their predicted probability of successful delivery of the correction message had they been in the social counter-partisan condition (principal stratification). Error bars reflect 95% confidence intervals.

The outcomes using this approach are qualitatively equivalent to our results when omitting the social counter-partisan condition. For overall engagement rate, we found a significant positive effect of social condition (b = 0.053, SE = 0.023, t(1447) = 2.31, p = 0.021; Fig 1), but did not find evidence of a significant effect of shared partisanship (b = 0.007, SE = 0.011, t(1447) = 0.57, p = 0.568; 95% credible interval = [-0.02, 0.03] using Bayesian regression analysis). We also did not find a significant interaction between conditions (b = 0.01, SE = 0.023, t(1447) = 0.45, p = 0.655; 95% credible interval = [-0.04, 0.06] ) on the probability of engagement with corrective message (See S75 and S82 Tables in S1 Text).

Our field experiment shows that a minimal social connection online can increase the likelihood of engagement with a correction message. We provide strong evidence that amongst co-partisans, social connections substantially increase engagement probability. Our results also suggest that this is also the case across co- and counter-partisans – though such evidence is less clear given differential treatment delivery failure rates between conditions. Interestingly, our results also show that shared partisanship is not a pre-requisite for correction engagement – we do not find a difference in engagement likelihood between co- and counter-partisans.

## Survey experiment 1

### Methods

To shed further light on the psychological effects of social connection and shared partisanship on engagement with social corrections, we conducted a follow-up survey experiment on MTurk. This survey allows us to (a) assess whether our field results replicate in a lab environment, (b) examine correction responses in a context that does not have the causal inference issues faced in the field experiment, and (c) preliminarily examine several potential mechanisms underlying the positive effect of minimal social connections on correction engagement.

**Participants.** Based on the results of a pilot study with a similar design (effect size $f^2 = .0125$), we performed a power calculation using G*Power [37] aiming for 90% statistical power, yielding a suggested total sample size of 843. We therefore aimed to recruit 850 participants via Amazon Mechanical Turk. As pre-registered, we also filtered for Twitter (X) users as an eligibility criteria. We recruited 815 participants who reported having a Twitter (X) account – 7 participants did not indicate partisanship, resulting in a final analysis of 808 participants (289 female, $M_{age} = 35.92$).

**Procedure.** Participants were asked to suppose they were on a social media platform such as Twitter. Given their randomly assigned treatment condition (baseline or social X counter-partisan or co-partisan), participants received a subset of the stimuli presented in Fig 2a and 2b. Corrector partisanship was indicated by candidate preference in the fictional account Twitter name.

Next, participants were instructed to suppose they received a public reply on their post (Fig 2b). Participants were then asked "Would you reply to the above message?", with response options "Yes, I would write a reply" and "No, I would not write a reply". Given that there was no actual post that the correction was responding to, we did not ask participants to write a reply. We then asked a variety of questions assessing potential mechanisms as to why participants may or may not have responded (see Sect 1c in S1 Text).

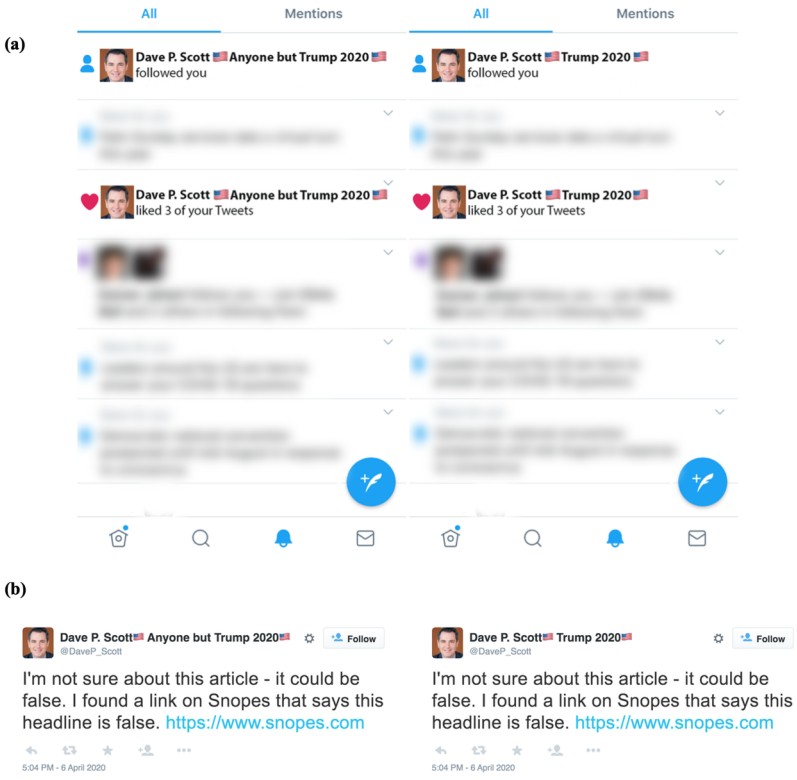

**Fig 2. Survey experiment 1 stimuli.** (a) Participants in the social condition viewed fictionalized Twitter notification screens, with corrections either from a Democrat user (left) or Republican user (right). Other notifications are blurred to focus the subject attention on the notification that is used as the manipulation. (b) Stylized corrective messages shown to participants (Democrat condition left; Republican condition right).

**Open science.** We pre-registered our experiment here. Deviations from our pre-registration are discussed in Sect 1a in S1 Text. All materials, data, and analysis code are available online here (https://osf.io/nvk4u/).

## Results

In our online survey experiment, as in our field experiment, we find a significant positive effect of prior social connection on the likelihood participants indicated they would reply to the correction (b = 0.083, SE = 0.035, t(800) = 2.373, p = .018). Also replicating our field results, we again surprisingly did not find evidence of a significant effect of shared partisanship on the probability of replying to the correction in a linear probability model (b = 0.028, SE = 0.017, t(800) = 1.608, p = .108; 95% credible interval = [-0.01, 0.06] using Bayesian regression analysis; see S2 and S32 Tables in S1 Text). Importantly, we also did not find a significant interaction between prior social connection and shared partisanship (b = 0.043, SE = 0.035, t(800) = 1.242, p = .214; 95% credible interval = [-0.03, 0.11] using Bayesian regression analysis).

These results were again replicated in a similar pre-registered online experiment (Supplemental Survey Experiment S1; N = 948) with a nearly identical design. Social connection significantly increased likelihood of engagement (b = 0.098, SE = 0.032, t(940) = 3.035, p = .002; see S42 Table in S1 Text), but we did not find a significant effect of shared partisanship on engagement (b = 0.050, SE = 0.032, t(940) = 1.533, p = .126; 95% credible interval = [-0.01, 0.11] using Bayesian regression analysis; see S61 Table in S1 Text) nor a significant interaction between social connection and shared partisanship (b = 0.025, SE = 0.065, t(940) = 0.393, p = .694; 95% credible interval = [-0.1, 0.15] using Bayesian regression analysis).

When pooling across both survey experiments, we find significant effects of both social connection (b = 0.093, SE = 0.024, t(1743) = 3.890, p < .001) and shared partisanship (b = 0.051, SE = 0.024, t(1743) = 2.137, p = .033; see S60 Table in S1 Text) on correction engagement. Though the effect of social connection is nominally greater, a follow-up linear hypothesis test does not provide evidence that the social connection and shared partisanship effects are significantly different in this pooled model (b = 0.042, SE = 0.034, t(1743) = 1.52, p = .218). We again do not find significant evidence for an interaction between social connection and shared partisanship on increasing correction engagement (b = 0.055, SE = 0.048, t(1743) = 1.151, p = .250; see Fig 3; S60 Table in S1 Text).

**Extreme partisanship moderates the effects of social connection and shared partisanship on correction engagement.** Given our surprising results of lack of strong evidence for an effect of shared partisanship on correction engagement in both our survey and field experiments, we also performed exploratory analyses examining the potential moderating role of partisan extremity. In Survey Experiment 1, we initially found that more politically extreme participants were more likely to engage with the correction (b = 0.048, SE = 0.017, t(800) = 2.733, p = .006; see S94 Table in S1 Text), but did not find evidence of interactions between participant partisan extremity and our treatment conditions (ps > .393; S94 Table in S1 Text).

However, we also investigated whether treatment effects may have differed for the most politically extreme users in our sample (i.e., 1 or 7 on 7-point partisanship measure; N = 222, 27.5% of total sample). In an exploratory, non-pre-registered analysis, we found a significant three-way interaction between social condition, shared partisanship, and an indicator variable for extreme partisanship (b = 0.214, SE = 0.078, t(800) = 2.743, p = .006; see Fig 4a, S97 Table in S1 Text). Decomposing this interaction, we found that social connection increased engagement (b = 0.101, SE = 0.041, t(582) = 2.471, p = .014; S95 Table in S1 Text) for lower-extremity

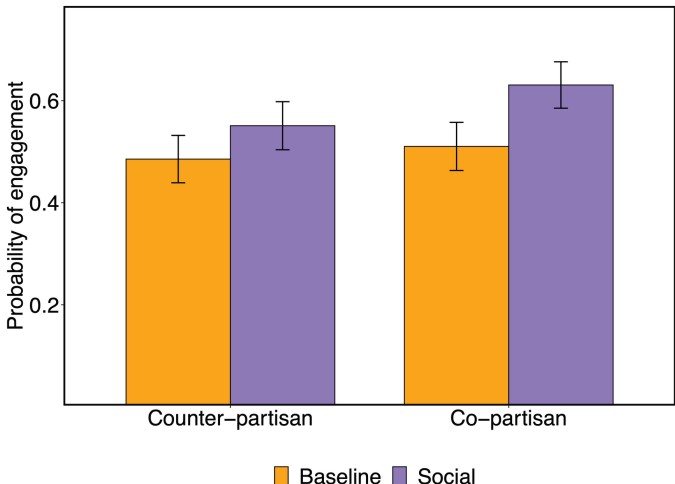

**Fig 3. Social connection increases engagement intentions in our survey experiments.** Likelihood of replying to corrective message by condition. Data pooled between main survey experiment and supplemental survey experiment S1. Error bars reflect 95% confidence intervals.

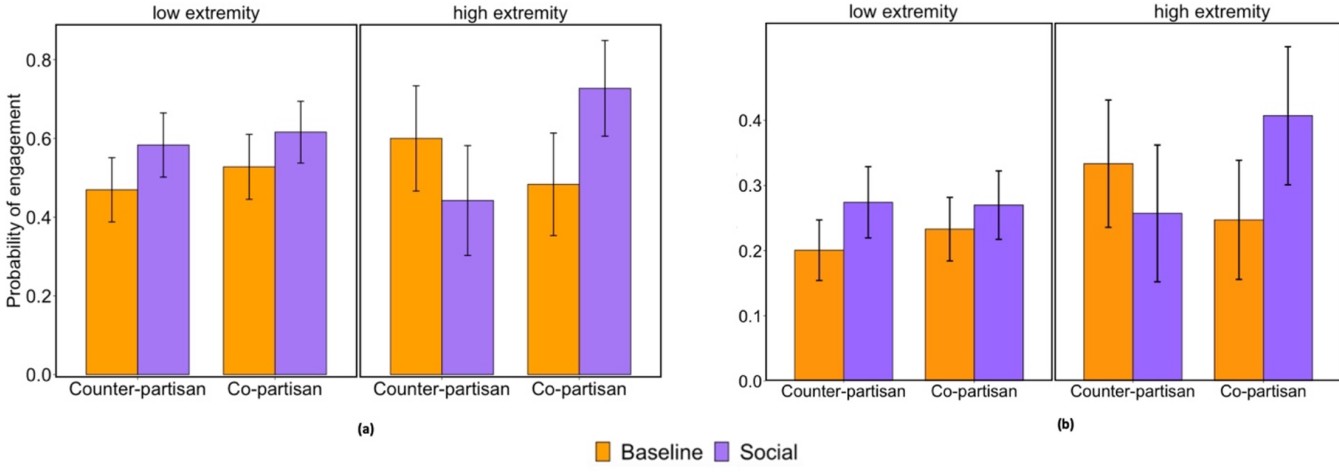

**Fig 4. Effect of prior social connection and shared partisanship on correction engagement, by partisan extremity.** (a) Survey experiment results. (b) Field experiment results – partisan extremity binned (low extremity = lowest 3 bins, high extremity = highest bin). Error bars reflect 95% confidence intervals.

participants without any effect or interaction with shared partisanship (ps > .268); but that prior social connection had no main effect on engagement for the most politically extreme participants. Rather, we find a significant interaction between social connection and shared partisanship (b = .201, SE = 0.066, t(218) = 3.062, p = .002; S96 Table in S1 Text) such that prior social connection increased engagement for co-partisans, but decreased engagement for counter-partisans.

We then revisited our field experiment data and conducted similar exploratory analyses. We initially do not find any significant interactions between user partisanship extremity, prior social connection, and/or shared partisanship (ps > .357; see S89 Table in S1 Text). However, we also investigated whether there may be non-linear effects of partisan extremity on engagement by experimental conditions. To match our survey experiment analyses, we classified

users into four bins based on partisan extremity (binned by inferred absolute value of partisanship [25] intervals of 0.625, from 0 to 2.5) and compared the most extreme users (highest bin: partisan score 1.88 to 2.5) to all other users (N = 355 high political extremity, 24.4% of total sample) in order to again compare the roughly quarter most politically extreme users to all other users. Replicating our survey findings, when omitting users for whom message delivery failed, we observe a significant three-way interaction (b = 0.136, SE = 0.054, t(1448) = 2.50, p = .012; see Fig 4b; S92 Table in S1 Text). This three-way interaction is also significant when including all users in our analysis, including those who never received a correction (b = 0.151, SE = 0.21, t(1578) = 2.99, p = .003; See S93 Table in S1 Text; S9 Fig in S1 Text). These results provide suggestive evidence that for less politically extreme users, social connection is beneficial overall, whereas for more politically extreme users, social connection is only beneficial among co-partisans – though we again note that these analyses examining high versus low politically extreme users are purely exploratory.

As pre-registered, we also examined potential mechanism-related dependent variables. We found suggestive evidence that participants in the social condition felt more obligated to reply to the corrective message (b = 0.300, SE = 0.147, t(800) = 2.031, p = .043; S4 Table in S1 Text), although this effect was not robust to adjusting for multiple comparisons given that we tested a variety of potential mechanistic variables (calculated q-value = 0.294; simulated q-value = 0.225; see Sect 1g in S1 Text). We did not observe any significant difference between the social and baseline conditions for any other potential mechanism variables (unadjusted ps > .138; see Sect 1c in S1 Text).

Together, our initial follow-up survey experiments largely replicate our field experiment results. Namely, we find that social connection increases engagement with social corrections. We find this to be the case for all less extreme partisans – though note that for extreme partisans, social connection increases engagement with co-partisan correctors but decreases engagement with counter-partisan correctors. We also provide some preliminary evidence for a mechanism – obligation to reply to messages from individuals who have even a minimal online connection.

## Survey experiment 2

### Methods

To follow up on this suggestive evidence, our next survey experiment again randomized participants into a social or non-social baseline condition. This time, we also randomized whether the message they received included a correction (as in the previous experiments) or was a non-correction placebo. The partisanship of the replying account was not indicated in this study. Participants were then randomly assigned to answer one of two key dependent variable items. Some participants were asked how obligated they felt to respond to the reply message; and other participants were asked the extent to which they believed other Twitter users would expect them to reply. Thus, the first dependent variable assesses personal obligation to respond, where the second assesses perception of a norm of engagement. We examine whether social connection increases obligations and perceptions of obligations to engage with corrections and non-corrections.

**Participants.** As pre-registered here, we recruited 1,606 participants on Amazon Mechanical Turk (644 female, $M_{age}$ = 36.90), screening for participants who reported having Twitter (X) accounts.

**Procedure.** Given their randomly assigned treatment condition (social, baseline), participants first received or did not receive stylized notifications of a human-appearing user following them and liking their past tweets on Twitter. Participants then either received

a correction message ("I'm not sure about this article – it could be false. I found a link on Snopes that says this headline is false. https://www.snopes.com.") or a non-correction message ("I'll check out this article – I'd like to know more. You might also find this story interesting: https://t.co/news-updates/...."). Participants next either answered a question on personal obligation to respond ("How obligated are you to respond to Dave P. Scott's reply?" 1 = Not at all obligated, 7 = Extremely obligated) or normative obligation to respond ("How much do you agree or disagree with the following statement? Most Twitter users would expect me to respond to Dave P. Scott's reply." 1 = Strongly Disagree, 7 = Strongly Agree).

**Open science.**   We pre-registered our experiment here. All materials, data, and analysis code are available online here (https://osf.io/nvk4u/).

**Ethics.**   Our survey experimental setups were exempted by the MIT Committee on the Use of Humans as Experimental Subjects (COUHES) exempt statuses #E-2443 and #E-1350. We received informed consent at the outset of all survey experiments.

## Results

We find via a general linear model that prior social connection significantly increased personal obligation to respond (b = 0.592, SE = 0.149, t(808) = 3.985, p < .001; see S35 Table in S1 Text, Fig 5a). We did not find a significant effect of correction condition on personal obligation to reply (b = 0.049, SE = 0.149, t(808) = 0.332, p = .740; 95% credible interval = [-0.24, 0.34] using Bayesian regression analysis; see S39 Table in S1 Text), nor did we observe a significant interaction between social condition and correction condition on feelings of personal obligation to reply (b = -0.154, SE = 0.297, t(808) = -0.518, p = .605; 95% credible interval = [-0.70, 0.42] using Bayesian regression analysis).

Similarly, we found that prior social connection significantly increased perceptions of a norm of responding (b = 0.245, SE = 0.121, t(790) = 2.019, p = .044; see S36 Table in S1 Text, Fig 5b). Additionally, we also found that receiving a corrective reply increased perceptions of a norm of responding relative to a non-corrective reply (b = 0.341, SE = 0.121, t(790) = 2.815,

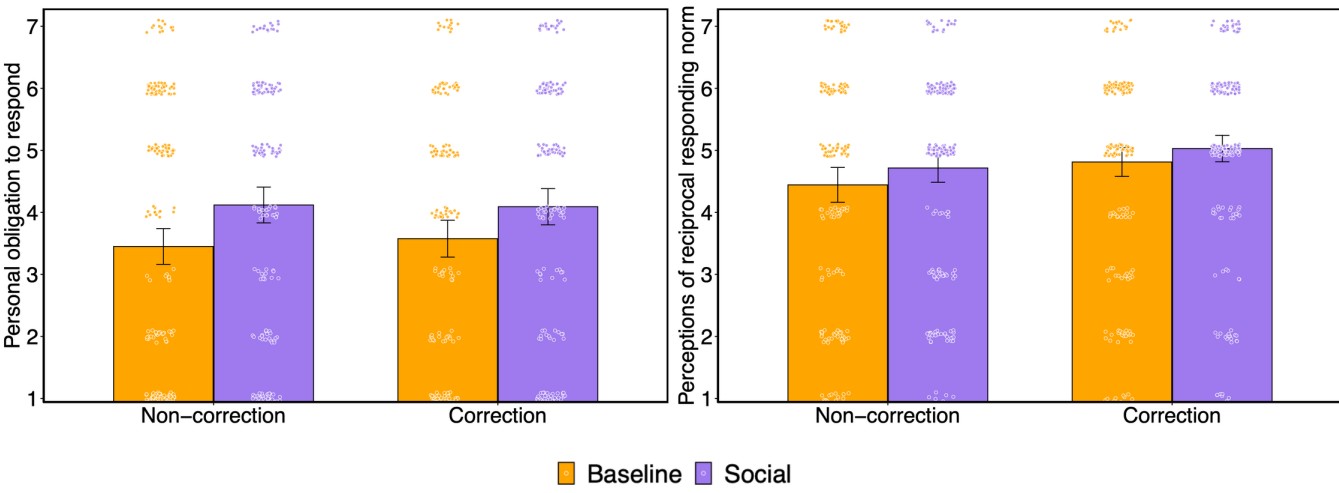

**Fig 5. Social connection increases reported personal obligation to reply and perceptions of a norm of reciprocal responding.** (a) Reported personal obligation to respond to messages by condition. (b) Reported perceptions of reciprocal responding norm by condition. Bars indicate means, and raw data are shown with dots. Error bars indicate 95% confidence intervals.

p = .005). However, we did not observe a significant interaction between social and correction conditions on perceived norms of responding (b = -0.056, SE = 0.242, t(790) = -0.230, p = .818; 95% credible interval = [-0.52, 0.40] using Bayesian regression analysis; see S40 Table in S1 Text).

In sum, our results support the idea that minimal social relationships increase feelings of obligation to respond, and increase perceptions of a social norm of responding. Furthermore, the results suggest that the effects of minimal social relationships on a communicative reciprocity norm are not unique to the domain of replying to corrections. Thus, there may be a more general social norm stipulating the need to respond to individuals based on prior social connections.

## Discussion

Across a Twitter (X) field experiment and follow-up online survey experiments, we provide evidence that "minimal" social connections online increase the probability of engagement with corrections, particularly among co-partisans. Our results also suggest that minimal social connections increase feelings of obligation to respond to public replies and increase perceptions of a general communicative reciprocity norm – even outside the context of replying to corrections. To our surprise, we did not find strong evidence that shared partisanship promotes engagement with corrections at baseline (i.e., in the absence of social connection). These results indicate the promise of minimal virtual social connections for dissuading users from simply ignoring corrections from co-partisans. Increasing correction engagement may have tangible benefits such as allowing those issuing social corrections greater direct feedback and insight into the immediate effects of their debunking attempts and providing the opportunity for longer-form engagement amongst users. That said, our field experiment results also show that social connection increased negative engagement – which may not be ideal for promoting belief updating via debunking.

The effect of social connection when the correction message came from a counter-partisan, however, was more complicated. In our field experiment, users in this condition were more likely to block our bots before correction message delivery – impairing our ability to assess whether, and how, they would have engaged with the correction had they received it. We do find via exploratory analyses that prior social connection may increase engagement with correction messages from counter-partisans for users who are not politically extreme – but for the most politically extreme users, social connection may decrease engagement with corrections from counter-partisans. This would be consistent with recent work on the importance of source credibility on correction efficacy – signaled counter-partisanship may incidentally discredit the correction source for extreme partisans [38]. It is important to note, however, that these perverse effects may be exacerbated in our experimental setting relative to typical engagement on Twitter (X) because our bots so strongly signaled their partisan commitments (e.g., referring to presidential candidates directly in their screen name). It seems plausible that social connection may effectively elicit greater overall engagement from counter-partisans when the corrector's partisanship is less strongly signaled (or when the corrector is politically neutral). Future work should investigate this possibility.

Our results on social connection provide potential guidance for effective and informative fact-check delivery on social media platforms – for instance, platform-directed correcting bots or user volunteers may also benefit from only correcting misinformation from users they have already recently followed or interacted with online; particularly if such bots express co-partisanship with users, or perhaps do not display partisan affiliation at all. Such prior interactions may then increase engagement with corrections of misinformation, which allows for

greater known feedback on corrections and allows for the possibility of extended engagement and dialogue with those sharing false claims.

Furthermore, recent advancements in large-language models have allowed for the development of user response prediction models designed to identify whether a social correction message will be beneficial neutral, or backfire [39]. A critical component of these models is training data consisting of existing responses to social corrections – and thus eliciting greater feedback and engagement with online corrections is integral for further developing such models and learning more about the features of corrections that predict corrective versus neutral or backfire effects. Our current work suggests that minimal contextual features such as establishing a minimal social connection via following soon-to-be corrected users can increase engagement with social corrections – thus allowing these models and researchers to learn more about how these corrections were received by users.

From a theoretical perspective, our second survey experiment also sheds light on a general communicative reciprocity norm that goes beyond just corrective messages. Given that we find that individuals in the social condition report a greater obligation to respond, and report that others would expect a response, it may be that there exists a general norm to reciprocate social contact on social media. Such a norm has been described, for example, in adolescent texting behavior [40]. Our experimental findings are also consistent with observational research on Twitter (X) reply networks which find that social relationships, rather than discussion topics, generally condition reply likelihood [41]. Given that we found these effects for both responding to corrective and non-corrective replies, our current findings reflect how a general communicative norm on Twitter (X) may be utilized for more effective and engaging social fact-checking.

Finally, social corrections are also beneficial via correcting the beliefs of third-party observers [42]. These benefits may be particularly acute for third-party users who also have existing social ties to the corrector – recent work suggests that individuals are more likely to re-share de-bunks from strong ties, as well as from correctors with shared partisanship [43]. Further, increased engagement with social corrections may allow for greater algorithmic amplification of the correction itself – which may further enhance the visibility and efficacy of the correction amongst third-party observers. Future research may examine whether engagement assists with the dissemination and visibility of corrections.

The current research has several limitations. We note multiple constraints on the generality of our findings. First, all bot accounts in our experiments were portrayed as White male accounts. Prior research has demonstrated differential responses to White- and Black-presenting accounts on Twitter (X) [29,44]. Future research should investigate the generalizability of our findings for non-White and non-male correctors. Second, our field experiment only targets users who shared one of 11 articles rated false by Snopes.com. Future work should investigate how our results generalize to other false articles – such as false news that has yet to be fact-checked by professional fact-checkers. Third, Mechanical Turk samples and Twitter (X) users who post links to debunked articles are highly non-representative of the American public. For instance, the majority of users in our field experiment are politically right-leaning – which, while non-representative of the general public, is reflective of empirical work showing that political conservatives are more likely to share and be exposed to low-quality content online [45–49]. And fourth, our field experiment and survey experiment 1 were conducted within the year leading up to the 2020 U.S. Presidential election. Thus, our results reflect the specific sociographic context of this time and location – the lead-up to a presidential contest within a polarized two-party system. Contemporary work may examine whether similar effects of social connection and shared partisan are observed under different political climates, cross-nationally, and in multi-party systems.

Relatedly, our results are limited in scope to examining social correction dynamics on Twitter (X). Different engagement norms may exist amongst various populations and on certain social media platforms. For instance, 'following' on Twitter (X) only necessitates a one-way social tie, whereas 'friending' on Facebook establishes a two-way tie. Likewise, blocking propensity and norms may differ across platforms, which may have implications for the efficacy of different bot designs (e.g., co-partisan versus counter-partisan social bots). Therefore, the extent to which prior social contact facilitates correction engagement, or engagement more generally, across platforms is an important direction for future research.

Our experiments are also limited by our selection and operationalization of minimal social connection and shared partisanship as corrector features of interest for our studies. First when establishing minimal social connections, in addition to initiating a unilateral social tie by following users on Twitter (X), we also liked three of users' tweets from the previous day in order to increase the salience and verisimilitude of our accounts. Future work may examine whether the extent of this minimal social contact my further impact the probability of users subsequently responding to corrective messages – for instance, liking a greater or lesser number of previous posts, or initiating more extensive contact such as commenting on other posts before issuing a corrective message. Second, given the time in which we conducted our field experiment, we operationalized account partisanship primarily through candidate preference. Given that candidate preference at the time we began our field experiment was highly indicative of partisanship [50], we posit that this is an effective way to manipulate shared or misaligned partisan preferences. That said, further research may examine how different representations expressing partisan preference (e.g., different candidate preferences; statements about party or issue position preferences) may differentially impact perceived shared partisanship and subsequent correction engagement. And finally, in the current work we focus on the corrector factors of partisan alignment and social connection between correctors and corrected users. Future work should examine whether other corrector or correction traits (e.g., if the correction receives likes or retweets from other third-party users) or relationships between correctors and corrected users (e.g., education level similarity or disparity between correctors and corrected users) may additionally impact correction engagement.

We also note that a follow-up analysis of the data reported in our paper suggests that social corrections may have unfortunate downstream consequences such as increasing the subsequent retweeting of low quality, toxic, or slanted content by the users who get corrected [11]. While such results may provide cause for concern regarding social corrections, such negative downstream effects may (i) be specific to low-attention sharing and not extend to belief updating per se, and (ii) may potentially be mitigated by continued conversation and discourse following the initial correction message. Indeed, such additional conversation would be the next step in correction approaches relying on discussion and argumentation. Thus, future research should examine how different types of social corrections and conversational approaches may best advance both engagement with, and efficacy of, corrections.

## Conclusion

In sum, we have shown that minimal pre-emptive social contact makes Twitter (X) users who post false news more likely to engage with corrections they receive, particularly from co-partisans. This engagement is not a proxy for efficacy – social contact notably increased initial negative engagement with correction messages. However, promoting such engagement, even if it is initially negative, allows for greater insight into the effects of social correction and may be an important first step for garnering more sustained dialogue and debunking efforts. It is easier to ignore corrective information from total strangers.

## Supporting information

**S1 Text**. Supporting information for "Promoting engagement with social fact-checks online: Investigating the roles of social connection and shared partisanship".
(PDF)

## Acknowledgments

Authors thank Ben Tappin for insightful comments on Bayesian analysis and Antonio Arechar for assistance with survey experiment data collection.

## Author contributions

**Conceptualization:** Cameron Martel, Mohsen Mosleh, Dean Eckles, David G. Rand.

**Data curation:** Cameron Martel, Mohsen Mosleh.

**Formal analysis:** Cameron Martel, Mohsen Mosleh, Dean Eckles.

**Funding acquisition:** David G. Rand.

**Investigation:** Cameron Martel, Mohsen Mosleh.

**Methodology:** Cameron Martel, Mohsen Mosleh, Dean Eckles.

**Project administration:** Cameron Martel, Mohsen Mosleh.

**Resources:** Cameron Martel, Mohsen Mosleh.

**Software:** Cameron Martel, Mohsen Mosleh.

**Supervision:** Dean Eckles, David G. Rand.

**Validation:** Cameron Martel.

**Visualization:** Cameron Martel, Mohsen Mosleh.

**Writing – original draft:** Cameron Martel, Mohsen Mosleh, David G. Rand.

**Writing – review & editing:** Cameron Martel, Mohsen Mosleh, David G. Rand.

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
