## [Decision Letter · Decision Letter 0]

13 Sep 2024

PONE-D-24-11037
Promoting engagement with social fact-checks online
PLOS ONE

Dear Dr. Mosleh,

Thank you for submitting your manuscript to PLOS ONE. After careful consideration, we feel that it has merit but does not fully meet PLOS ONE’s publication criteria as it currently stands. Therefore, we invite you to submit a revised version of the manuscript that addresses the points raised during the review process.

We look forward to receiving your revised manuscript.

Kind regards,

Stefano Cresci

Academic Editor

PLOS ONE

 [The authors gratefully acknowledge funding from Google through Google Research Scholar Award, gifts from Google to support other research, and funding via the National Science Foundation Graduate Research Fellowship under Grant No. 174530.]. 

[Other research by the authors was funded by Meta and Google.].

[Authors thank Ben Tappin for insightful comments on Bayesian analysis and Antonio Arechar for assistance with survey experiment data collection. The authors gratefully acknowledge funding from Google through Google Research Scholar Award, gifts from Google to support other research, and funding via the National Science Foundation Graduate Research Fellowship under Grant No. 174530.]

[The authors gratefully acknowledge funding from Google through Google Research Scholar Award, gifts from Google to support other research, and funding via the National Science Foundation Graduate Research Fellowship under Grant No. 174530.]. 

6. We note that you have indicated that there are restrictions to data sharing for this study. PLOS only allows data to be available upon request if there are legal or ethical restrictions on sharing data publicly. For more information on unacceptable data access restrictions, please see http://journals.plos.org/plosone/s/data-availability#loc-unacceptable-data-access-restrictions.

Additional Editor Comments:

Both reviewers found merit in this manuscript, however both also identified a number of relevant issues that must be addressed for the paper to be published. Among the issues requiring particular attention are the methodological choices which require better justification and description, as well as improvements in certain areas; and some aspects of the study design that need stronger justification or the adoption of alternative approaches to avoid possible biases. A reviewer also commented that the provided supplementary materials and code were insufficiently documented and unclear. Based on the somewhat contrasting feedback that I received, I opted for an R&R decision. However, I foresee that thoroughly addressing the raised issues might be challenging. I encourage the authors to carefully consider the reviewers' reports and assess whether they are able to address the majority of the comments in a revised version of the work.

Reviewers' comments:

Reviewer's Responses to Questions

**Comments to the Author**

1. Is the manuscript technically sound, and do the data support the conclusions?

Reviewer #1: Yes

Reviewer #2: Partly

2. Has the statistical analysis been performed appropriately and rigorously? 

Reviewer #1: Yes

Reviewer #2: Yes

3. Have the authors made all data underlying the findings in their manuscript fully available?

Reviewer #1: No

Reviewer #2: Yes

4. Is the manuscript presented in an intelligible fashion and written in standard English?

Reviewer #1: Yes

Reviewer #2: Yes

5. Review Comments to the Author

Reviewer #1: The authors report that the data will be available although "some restrictions will apply." These restrictions are not outlined.

I have a few minor comments on the manuscript as follows:

- Page 3: "Our experiments focus on two factors which prior work suggest may be important for eliciting engagement with corrections: partisan alignment between the corrector and the corrected user, and social connection between the corrector and the corrected user." -- are there other factors the authors think would be relevant to explore in future research aside from these 2? For example, one could study whether the correction post came from a highly educated individual or not, or if the correction post was engaged with (liked, retweeted) or not (and if so, by whom).

- Page 4: "What characteristics of correction senders promote engagement with social corrections?" -- is this meant to be the research question motivating this study? This sentence does not flow well with the paragraph that follows given that the authors are studying 2 very specific characteristics (partisanship and social connection). In other words, the authors aren't answering the question they pose.

- Page 4: "We varied whether our corrector bots identified as Democrats or Republicans, and whether they followed the user and liked three of their tweets the previous day (thus creating a minimal social relationship)." -- did the authors come up with this definition of "minimal social relationship"? Why 3 likes instead of 2 or 10? This seems arbitrary, and is not justified in the current version of the manuscript. This plays a rather large role in the framing/design of the studies and needs to be clarified.

- For the field experiment, how was the Twitter data collected?

-Footnote 1 on page 6: The information there is highly redundant as the 2 deviations are also reported on page 5. Suggest removing redundant content.

- The methods of analysis are not clearly motivated or explained anywhere in the paper. For example, the authors use a random forest model on Page 7. Why? The reader is never brought through the methodological choice. Bayesian regression and GLMs are used at other points in the study. Again, why? These methodological components should be clearly introduced before being deployed.

Reviewer #2: The manuscript concerns peer-based correction of misinformation related to United States (US) politics, namely the potential influence of the corrector’s partisanship and a prior (minimal) online social tie on the engagement of the corrected user. The manuscript presents three closely related experiments —one field experiment on Twitter (now X) and two surveys on MTurk— which analyzed different aspects of these two traits. Based on their results, Authors found that a shared partisanship does not have a significant effect when there is no prior online tie, but when this tie exists, shared partisanship increases engagement (more so for negative replies).

In general, I found the core inquiry of the manuscript compelling, i.e., how certain traits of a misinformation corrector influence engagement of the corrected user. This is, in my view, an aspect of general interest for social media researchers and experts in general, not only scholars working on misinformation. In addition, I greatly appreciated the fact that Authors pre-registered their experiments (indicating deviations when appropriate), got approval by their internal review board, and released the code and data they used for their analyses.

Nevertheless, I found several issues in the current state of the manuscript (and supplemental material) that together greatly hinder its publication. These issues range from framing of the study and construction of the dataset and bot accounts to arguably unsuitable statistical analysis and presentation of results. Given that some of these concerns are related to the way in which the data was collected and analyzed, these might represent a considerable problem for the conclusions presented in the manuscript.

Regarding the framing of the study, I believe that the title and abstract should be more specific with respect to the work carried out. For instance, in the abstract it is stated that “we ask what types of social correction promote engagement from corrected users”, which initially led me to think that the study focuses on variations of the correction themselves. However, the study only focuses on two traits between corrector and corrected users: partisanship and online social tie, with the latter being narrowed down to the corrector following the corrected user, i.e., a unidirectional and unilateral tie. In addition, I think the sociogeographic context of the study (the US political ecosystem a few months prior to the 2020 US elections) should be clearly stated and described, as this might have influenced the results and might not be obvious for readers not familiar with US politics. Hence, I believe the discussion and conclusions should be framed more narrowly.

Regarding the construction of the fact-checking dataset and bot accounts for the field experiment, I found some balance issues that might have affected results. In particular, there is —in my view— a noticeable imbalance in the estimated liberal–conservative partisanship scale (measured from -2.5 to +2.5) of the eleven fact-check URLs of Snopes —§4.a of the Supplemental Information (SI). As can be seen on Table 62 of SI, seven of the eleven Snopes’ URLs have a marked conservative estimated partisanship (ranging from +1.19 to +1.74), whereas the remaining four liberal URLs are more center leaning (ranging from -0.27 to -1.1). Perhaps, instead of taking the absolute most recent articles, it would have been better to filter them until obtaining a better partisanship balance. In addition, in the Procedure paragraph of Survey Experiment 1, it is stated that “corrector partisanship was indicated by candidate preference in the fictional account Twitter name”, with the caption of Figure 2.b indicating “Democrat condition left; Republican condition right”. However, the respective usernames read “Anyone but Trump 2020” and “Trump 2020”, which I would argue it’s only reflective of partisanship in favor or against Trump, but not necessarily towards a political party. In my view, the respective username statements are not equivalent and might have introduced bias against the so-called Democrat condition.

Regarding the statistical analyses, I think that while Authors tried to be exhaustive in their tests and results in the SI —with almost 100 supplementary tables (too many in my opinion) and 8 supplementary figures— there are some crucial issues in the manuscript itself. There are small reference errors, such as in p. 5 in which it is stated that the 12 variations of the corrective messages are in SI Table 59, but these are in SI Table 63. And, above all, there are decisions that are not well explained. For instance, I did not understand the binning process for extreme partisanship in Survey Experiment 1. Does it refer to quartiles given that there are four bins? In that case, I’m not sure if the highest quartile could be called extreme partisanship, with the analyses based on this binning seemingly being too arbitrary. The rationale to conduct certain analyses and tests is not given, particularly the use of both Bayesian and non-Bayesian regression. In addition, the analyses of Likert data are done treating these as interval data, whereas I would argue that it is more appropriate to treat them as ordinal (Göb et al., 2007), albeit this is open to debate. In either case, I think that the graphical representation in Fig. 5 is not suitable for the Likert data at hand; I suggest using instead a diverging stacked bar chart. For the R ecosystem, there are the Likert-related functions of the “HH” package or, if using “ggplot”, the package “likert”.

Speaking of R, while I really appreciate that Authors have published the code used for the analyses, in my opinion the way it is structured and not documented greatly hinders reproducibility, and casts doubts on the way the analyses were conducted. For instance, there is no README to guide the interested reader throughout the different files or the libraries needed (and why). Among several things, it is cumbersome to navigate the files and code; it is difficult to discern what is exploratory and what is not; what is the order of the analyses; some notebooks have a few comments explaining a given step but others not at all. For instance, in “SnopeFieldReplies.Rmd” there is a section in which several values are manually assigned, for which “Fix NAs” is the only comment; why are these values missing and how were they recovered? There is also the use of the function “setwd” in several Rmd notebooks that change the working directory to a location present only within the workstation of the original analyst, instead of using a relative path that should work across different workstations.

On a different note, an important difficulty and aspect of the field experiment is the blocking of some of the bot accounts by targeted counter-partisan users, which as noted by Authors, impairs the “ability to assess whether, and how, they would have engaged with the correction had they received it”. I found this phenomenon to be interesting and worth of further investigation to improve the current manuscript. However, I noticed that three of the four Authors have already recently published a research report on this very topic (Martel et al., 2024), which I found interestig and with a more straightforward approach compared to the current manuscript. This other paper, however, brought to my attention that the cited literature in the current manuscript is not the most recent on the topic, with the year 2022 being the latest one (3 out of 33 references).

Given all of the above, I think that a thorough revision is warranted across the manuscript and the supplemental material. I am, however, afraid that this might go beyond a major revision due to the aforementioned issies collection of data and analyses.

REFERENCES:

• Göb, R., McCollin, C., & Ramalhoto, M. F. (2007). Ordinal methodology in the analysis of Likert scales. Quality & Quantity, 41(5), 601-626.

• Martel, C., Mosleh, M., Yang, Q., Zaman, T., & Rand, D. G. (2024). Blocking of counter-partisan accounts drives political assortment on Twitter. PNAS nexus, 3(5), page 161.

6. PLOS authors have the option to publish the peer review history of their article (what does this mean?). If published, this will include your full peer review and any attached files.

Reviewer #1: No

Reviewer #2: No

---

## [Author Response · Author response to Decision Letter 1]

9 Jan 2025

Please see uploaded response letter in pdf

---

## [Decision Letter · Decision Letter 1]

31 Jan 2025

Promoting engagement with social fact-checks online:

Investigating the roles of social connection and shared partisanship

PONE-D-24-11037R1

Dear Dr. Martel,

We’re pleased to inform you that your manuscript has been judged scientifically suitable for publication and will be formally accepted for publication once it meets all outstanding technical requirements.

Kind regards,

Chang Sup Park, Ph.D.

Academic Editor

PLOS ONE

Additional Editor Comments (optional):

In light of the reviewers' comments, we are glad to accept your paper.

Reviewers' comments:

Reviewer's Responses to Questions

**Comments to the Author**

1. If the authors have adequately addressed your comments raised in a previous round of review and you feel that this manuscript is now acceptable for publication, you may indicate that here to bypass the “Comments to the Author” section, enter your conflict of interest statement in the “Confidential to Editor” section, and submit your "Accept" recommendation.

Reviewer #1: All comments have been addressed

Reviewer #2: All comments have been addressed

2. Is the manuscript technically sound, and do the data support the conclusions?

Reviewer #1: Yes

Reviewer #2: Yes

3. Has the statistical analysis been performed appropriately and rigorously? 

Reviewer #1: Yes

Reviewer #2: Yes

4. Have the authors made all data underlying the findings in their manuscript fully available?

Reviewer #1: Yes

Reviewer #2: Yes

5. Is the manuscript presented in an intelligible fashion and written in standard English?

Reviewer #1: Yes

Reviewer #2: Yes

6. Review Comments to the Author

Reviewer #1: (No Response)

Reviewer #2: I really appreciate that Authors have addressed all my concerns —as well as those of Reviewer 1— and have performed a major revision of their work and supplemental material. I believe that these changes greatly improve the manuscript and ease its reproducibility. Indeed, with a narrower framing, clearer methodology and more explicit limitations, I think that the manuscript’s results and conclusions hold better to scrutiny. If I were to nitpick, I would say that the number of tables in the Supplemental Information remains too high for the work at hand, given the availability of data and code. In this case, I would say that comprehensiveness is preferable over exhaustiveness.

7. PLOS authors have the option to publish the peer review history of their article (what does this mean?). If published, this will include your full peer review and any attached files.

Reviewer #1: No

Reviewer #2: No

---

## [Editor Report · Acceptance letter]

PONE-D-24-11037R1

PLOS ONE

Dear Dr. Martel,

I'm pleased to inform you that your manuscript has been deemed suitable for publication in PLOS ONE. Congratulations! Your manuscript is now being handed over to our production team.

Kind regards,

on behalf of

Dr. Chang Sup Park

Academic Editor

PLOS ONE